# Extra Virgin Olive Oil Contains a Phenolic Inhibitor of the Histone Demethylase LSD1/KDM1A

**DOI:** 10.3390/nu11071656

**Published:** 2019-07-19

**Authors:** Elisabet Cuyàs, Juan Gumuzio, Jesús Lozano-Sánchez, David Carreras, Sara Verdura, Laura Llorach-Parés, Melchor Sanchez-Martinez, Elisabet Selga, Guillermo J. Pérez, Fabiana S. Scornik, Ramon Brugada, Joaquim Bosch-Barrera, Antonio Segura-Carretero, Ángel G. Martin, José Antonio Encinar, Javier A. Menendez

**Affiliations:** 1ProCURE (Program Against Cancer Therapeutic Resistance), Metabolism & Cancer Group, Catalan Institute of Oncology, 17007 Girona, Spain; 2Girona Biomedical Research Institute (IDIBGI), 17190 Girona, Spain; 3StemTek Therapeutics, 48160 Bilbao, Spain; 4Department of Analytical Chemistry, Faculty of Sciences, University of Granada, 18071 Granada, Spain; 5Research and Development Functional Food Centre (CIDAF), PTS Granada, 18100 Granada, Spain; 6Cardiovascular Genetics Centre, Department of Medical Sciences, University of Girona, 17071 Girona, Spain; 7Mind the Byte, 08007 Barcelona, Spain; 8Centro de Investigación Biomédica en Red de Enfermedades Cardiovasculares (CIBERCV), 28029 Madrid, Spain; 9Faculty of Medicine, University of Vic-Central University of Catalonia (UVic-UCC), 08500 Vic, Spain; 10Dr. Josep Trueta Hospital of Girona, 17007 Girona, Spain; 11Medical Oncology, Catalan Institute of Oncology (ICO), 17007 Girona, Spain; 12Department of Medical Sciences, Medical School University of Girona, 17071 Girona, Spain; 13Institute of Research, Development and Innovation in Biotechnology of Elche (IDiBE) and Molecular and Cell Biology Institute (IBMC), Miguel Hernández University (UMH), 03202 Elche, Spain

**Keywords:** phenolics, secoiridoids, cancer, cancer stem cells, SOX2, metabolism, neurological disorders

## Abstract

The lysine-specific histone demethylase 1A (LSD1) also known as lysine (K)-specific demethylase 1A (KDM1A) is a central epigenetic regulator of metabolic reprogramming in obesity-associated diseases, neurological disorders, and cancer. Here, we evaluated the ability of oleacein, a biophenol secoiridoid naturally present in extra virgin olive oil (EVOO), to target LSD1. Molecular docking and dynamic simulation approaches revealed that oleacein could target the binding site of the LSD1 cofactor flavin adenosine dinucleotide with high affinity and at low concentrations. At higher concentrations, oleacein was predicted to target the interaction of LSD1 with histone H3 and the LSD1 co-repressor (RCOR1/CoREST), likely disturbing the anchorage of LSD1 to chromatin. AlphaScreen-based in vitro assays confirmed the ability of oleacein to act as a direct inhibitor of recombinant LSD1, with an IC_50_ as low as 2.5 μmol/L. Further, oleacein fully suppressed the expression of the transcription factor SOX2 (SEX determining Region Y-box 2) in cancer stem-like and induced pluripotent stem (iPS) cells, which specifically occurs under the control of an LSD1-targeted distal enhancer. Conversely, oleacein failed to modify ectopic SOX2 overexpression driven by a constitutive promoter. Overall, our findings provide the first evidence that EVOO contains a naturally occurring phenolic inhibitor of LSD1, and support the use of oleacein as a template to design new secoiridoid-based LSD1 inhibitors.

## 1. Introduction

Natural biophenols in extra virgin olive oil (EVOO), a health-promoting ingredient of the so-called Mediterranean Diet, have been proposed to have multifaceted and beneficial effects on major diseases, including cancer, diabetes, cardiovascular diseases, neurodegenerative disease, and metabolic disorders [1,2,3,4,5,6,7,8,9]. Although the mechanism(s) of action of EVOO biophenols in these settings is not yet fully understood, there is emerging evidence suggesting that phenolic molecules may exert nutritional and beneficial actions through multiple molecular mechanisms involving not only general biochemical pathways but also specific epigenetic remodeling [10,11].

We have recently used a reverse pharmacology computational approach to “de-orphanize” endogenous ligands and assign molecular functions to the biophenol oleacein [12]—one of the most abundant secoiridoids in EVOO that seems to be largely responsible for its health-promoting effects [6]. Our virtual profiling approach employing structure-based tools revealed a target landscape of oleacein significantly enriched in metabolic and chromatin-modifying enzymes involved in histone post-translational modification. Among the latter group was lysine-specific demethylase-1 (LSD1, also known as KDM1A), a flavin adenine dinucleotide (FAD)-dependent homolog of the amine oxidase family that was originally described to target histone H3K4 [13,14]. LSD1 has been linked to a repression of gene expression, and was later reported to interact with the androgen or estrogen receptor to demethylate histone H3K9 [15,16,17], resulting in the activation of gene expression.

LSD1 plays a central epigenetic role in nutrient-driven metabolic adaptation and reprogramming in multifactorial diseases, including obesity-associated diseases, neurological disorders, and cancer [18,19,20,21,22,23,24,25]. Accordingly, natural and synthetic chemicals capable of blocking LSD1 might represent a new class of epigenetic drugs to treat a wide range of metabolic disorders. In this regard, some of the beneficial effects promoted by polyphenols, such as resveratrol, curcumin, quercetin, and baicalin, could be explained, at least in part, by their ability to operate as natural inhibitors of LSD1 [26,27,28,29]. Here, we combined in silico docking and molecular dynamics approaches coupled to experimental validation to identify LSD1 as a novel epigenetic target of the bioactive EVOO polyphenol oleacein.

## 2. Materials and Methods

### 2.1. Docking Calculations, Molecular Dynamics Simulations, and Binding Free Energy Analysis

The human histone demethylase LSD1 (UniProt code O60341)/REST corepressor 1 (UniProt code Q9UKL0)/histone H3 peptide (UniProt code P68431) ternary complex structures (PDB codes 2IW5 and 2X0L) were obtained from the Research Collaboratory for Structural Bioinformatics Protein Data Bank (PDB). The structure of oleacein (PubChem CID: 18684078) was obtained from the National Center for Biotechnology Information (NCBI) PubChem database (http://www.ncbi.nlm.nih.gov/pccompound). The specific edition of the LSD1 protein structure involving the removal of water, FAD, and histone H3 peptide, was made using PyMol 2.0 software (PyMOL Molecular Graphics System, v2.0 Schrödinger, LLC, at http://www.pymol.org/) without further optimization.

Docking calculations, molecular dynamics (MDs) simulations, and molecular mechanics/generalized borne surface area (MM/GBSA) calculations to determine the alchemical binding free energy of oleacein against LSD1 2IW5 were performed as previously described [10,11,12,30]. Docking calculations, MD simulations, and molecular mechanics Poisson–Boltzmann surface area (MM-PBSA) calculations to determine the alchemical binding free energy of oleacein against LSD1 2X0L were performed as previously described [31]. All of the figures were prepared using PyMol 2.0 software and all interactions were detected using the protein-ligand interaction profiler (PLIP) algorithm [32].

### 2.2. Oleacein Isolation and Purification

Oleacein (decarboxymethyl oleuropein aglycone) was isolated and purified from the phenolic fraction of EVOO as previously described [10].

### 2.3. LSD1 Enzymatic Activity

Enzymatic reactions were performed in an AlphaScreen format in duplicate at room temperature for 60 min in a 10 μL mixture containing assay buffer, histone H3 peptide substrate, LSD1 (BPS#50103, lot#130806-D) enzyme, and oleacein. The 10-μL reactions were carried out in 384-well Optiplates (Perkin Elmer Life Sciences, Waltham, MA). A serial dilution of the compounds was first performed in 3.3% dimethylsulfoxide (DMSO)/assay buffer. From this step, 3 μL of oleacein was added to 4 μL of enzyme and incubated for 30 min at room temperature. After this incubation, 3 μL of substrate was added to initiate the reaction. The final DMSO concentration was 1%. After the reaction, 5 μL of anti-mouse acceptor beads (Perkin Elmer, diluted 1:500 with 1× detection buffer) or 5 μL of anti-rabbit acceptor beads (Perkin Elmer, diluted 1:500 with 1× detection buffer) and 5 μL of primary antibody (BPS#52140E,F, diluted 1:200 with 1× detection buffer) were added to the reaction mix. After brief shaking, the plate was incubated for 30 min. Finally, 10 μL of AlphaScreen streptavidin-conjugated donor beads (Perkin Elmer, diluted 1:125 with 1× detection buffer) were added. After 30 min, the samples were measured on an AlphaScreen microplate reader (EnSpire Alpha 2390 Multilabel reader, Perkin Elmer).

The AlphaScreen intensity data were analyzed and compared using Graphpad Prism software (GraphPad Software Inc., San Diego, CA, USA). In the absence of iadademstat, the AlphaScreen or fluorescence intensity (F_t_) was defined as 100% activity. In the absence of enzyme, the intensity (F_b_) was defined as 0% activity. The percent activity in the presence of oleacein was calculated according to the following equation: %activity = (F − F_b_)/(F_t_ − F_b_), where F = the A-screen intensity in the presence of iadademstat. Once A-screen data were converted to LSD1 activity (%), those values were then plotted against a series of oleacein concentrations using non-linear regression analysis of sigmoidal dose–response curves generated with the equation Y = B + (T − B)/1 + 10^((LogIC50-Z) × HillSlope), where Y = percent activity, B = minimum percent activity, T = maximum percent activity, Z = logarithm of oleacein concentration, and Hill Slope = slope factor or Hill coefficient. The IC_50_ value was determined as the concentration of oleacein causing a half-maximal inhibition of control activity.

### 2.4. Cell Lines and Culture Conditions

MCF-7 and BT-474 breast carcinoma cell lines were obtained directly from the ATCC (Manassas, VA, USA) and grown in Dulbecco’s modified Eagle’s medium (Gibco, Carlsbad, CA, USA) supplemented with 10% fetal bovine serum (Sigma) at 37 °C in a 5% CO_2_ incubator. MCF-7 cells engineered to stably overexpress SOX2 were prepared and cultured as previously described [33]. Reprogramming of human fibroblasts from a healthy individual to induced pluripotent stem (iPS) cells was carried out as previously described [34].

### 2.5. SOX2 Enhancer Reporter Assay

BT-474 cells were transfected with 5 μg of pGL3 Luc control (Promega, Madison, WI, USA) or pGL2-Sox2-enhancer-Luc reporter plasmids [34,35,36] using Lipofectamine Plus (Invitrogen, Carslbad, CA, USA). Twenty-four hours after transfection, the culture was split into two parts: One part was seeded in two-dimensional adherent culture plates and the other part was cultured in non-adherent culture conditions to allow mammosphere formation, in the absence or presence of graded concentrations of oleacein (i.e., 0, 3, 10, and 30 μmol/L). After 48 h, cells were harvested and luciferase activity was measured in duplicate with the Glomax 20/20 luminometer (Promega) and normalized by protein concentration in the extracts. Results are expressed as the fold induction of sphere culture reporter activity above adherent culture control.

### 2.6. SOX2 Protein Expression

To evaluate the expression status of SOX2, cells grown to 80% to 90% confluence were exposed to graded concentrations of oleacein (i.e., 0, 5, 10, 25, and 50 μmol/L) for 24 h. Cells were then collected and subjected to western blot analysis with a monoclonal antibody against SOX2 (Cat.# 3579, Cell Signaling). β-actin (Cat.# 66009-1-Ig, Clone No. 2D4H5) was used as loading control.

### 2.7. SOX2 Gene Expression

Total RNA was extracted from cells using a Nucleospin RNA plus kit (Macherey-Nagel GmbH & Co. KG Düren, Germany). One microgram of total RNA was reverse-transcribed into cDNA using the High Capacity cDNA Reverse Transcription Kit (Thermo Fisher Scientific, Carlsbad, CA). RNA concentration and quality were determined with an ND-1000 spectrophotometer (NanoDrop™ ND-1000, NanoDrop Technologies, Wilmington, DE).

cDNA (20 ng) was assayed in triplicate according to established protocols using a QuantStudio™ 7 Flex Real-Time PCR system (Thermo Fisher Scientific, Waltham, MA) with an automated baseline and threshold cycle detection. *PPIA* and *18S* were used as reference genes. Primers and fluorescent probes for *PPIA*, *18S*, and *SOX2* were obtained from Thermo Fisher Scientific (TaqMan Gene Expression assay IDs: Hs99999904_m1, Hs99999901_s1, and Hs01053049_s1, respectively). Data were analyzed using the Thermo Fisher Cloud software.

### 2.8. Statistical Analysis

All statistical analyses were performed using GraphPad Prism software. Data are presented as mean ± S.D. Comparisons of means of ≥3 groups were performed by analysis of variance (ANOVA) and the existence of individual differences, in case of significant *F* values at ANOVA, were assessed by multiple contrasts. *p* values < 0.05 were considered to be statistically significant (denoted as *). All statistical tests were two-sided.

## 3. Results

### 3.1. Computational Prediction of LSD1 as a Putative Target of Oleacein

One of the epigenetic targets predicted to be engaged by oleacein in our recent chemoinformatics approach [12] was PDB ID 2IW5 [37], which corresponds to a complex containing LSD1 (KDM1A, Uniprot O60341) and REST Corepressor 1 (RCOR1/CoREST), a co-repressor that cooperates with LSD1 to demethylate mono- and di-methylated lysine 4 (K4) of histone H3 in nucleosomes. A classic rigid docking calculation, which was performed twice to avoid false positives, over the structure employing the cavity defined by its crystallographic ligand (FAD) revealed an energy binding of −7.2/−7.5 kcal/mol (Table 1; Figure 1A, top). Blind docking calculations involving cavity searching and docking calculations over the established cavities confirmed that the best predicted cavity for oleacein in LSD1 was the same as that containing FAD (−6.1 kcal/mol; Table 1; Figure 1A, bottom). To add protein flexibility to the analysis and to test the stability of the predicted oleacein-LSD1 complex, we performed short molecular dynamics (MDs) simulations of 1 ns to filter out the possibility that the predicted LSD1–oleacein complex would involve a poorly interacting pose (Figure 1B). We then performed a molecular mechanics/generalized born surface area (MM/GBSA) approximation of the oleacein–LSD1 binding affinity, which considers the dynamic nature of LSD1 and therefore provides a more realistic view of oleacein binding affinity than the aforementioned rigid rocking estimations. The results highlighted a high affinity of oleacein at the FAD cavity of LSD1, which reached −39.9541 kcal/mol.

When assessing the most relevant interactions between oleacein and LSD1 and the residues involved in such binding when placed over the FAD cavity, it was predicted that among the residues involved in the FAD-binding mode (K661, Y761, A809, and T801), two of them (K661 and Y761) were shared by the oleacein-binding mode. The stability of oleacein in the FAD-binding pocket was predicted to involve hydrogen bond interactions with K661, A331, and M332, as well as hydrophobic contacts with R316, V333, F538, and V811. Oleacein was predicted to establish ∏-cation interactions with W751. Beyond a predicted LSD1 inhibitory mode of oleacein via occupancy of the FAD site, it is noteworthy that the oleacein interacting residues F538 and V811 are placed near the residues involved in the binding of histone H3 (N540, P808, and A809), thereby suggesting that the inhibitory capacity of oleacein to LSD1 could also involve a disturbance of the anchorage of LSD1 to chromatin.

### 3.2. Computational Deconstruction of the Binding Mode of Oleacein to the LSD1-CoREST-histone H3 Complex

To computationally validate the virtual profiling prediction with 2IW5, we re-evaluated the binding mode of oleacein to the LSD1-CoREST-histone H3 complex using a different crystal structure—namely the 2X0L ternary complex structure [38]. Docking simulations of oleacein in such a crystal structure of human LSD1 (chain A), including RCOR1/CoREST (chain B), and a histone H3 peptide (chain C), produced eight clusters of docking poses (Figure 2).

When the docking results were ranked according to the ascent of the binding energies for oleacein (up to −9.63 kcal/mol; Table 2), the clusters exhibiting the highest affinity (#1 and #2) were in the nanomolar range and were predicted to interact with those residues delineating the FAD cavity of LSD1. This was the case particularly for cluster #2 (−9.58 kcal/mol), in which all four residues involved in the FAD-binding mode (K661, Y761, A809, and T801) were shared by the oleacein-binding mode. Cluster #3 (−9.33 kcal/mol), in the sub-micromolar range, was predicted to interfere with the anchorage of LSD1 to chromatin as it included two (P808 and A809) out of the three residues involved in the binding of histone H3. Clusters #7 and #8, with affinities in the low micromolar range, were predicted to interact with both the LSD1 enzyme and RCOR1/CoREST.

To add protein flexibility and provide additional information about different intra- and inter-molecular movements, we performed short MD simulations over the course of 10 ns together with binding free energy calculations under the molecular mechanics Poisson–Boltzmann surface area (MM-PBSA) approximation (Table 2). With the exception of clusters #4 and #5, the MM-PBSA-based results highlighted the predicted extremely high affinity of oleacein for the FAD-binding pocket of LSD1 at the cluster #2, which reached −155.861 kcal/mol. For the remaining clusters, the range of binding energies was notably high (up to −66.844 kcal/mol in cluster #1), with oleacein trajectory changes of less than 4 Å (data not shown). The high binding energies of clusters #7 and #8 (−40.682 and −30.051kcal/mol, respectively), with affinities in the low micromolar range, are especially relevant as they predict that oleacein might stably bind and obstruct the interaction between LSD1 and RCOR1/CoREST, likely disturbing the anchorage of LSD1 to chromatin.

### 3.3. Oleacein Inhibits the In Vitro Enzymatic Activity of LSD1

Using an AlphaScreen assay with a biotinylated histone H3 peptide substrate, purified LSD1, and a highly specific antibody that recognizes demethylated substrate, we experimentally confirmed the model predictions of oleacein to target LSD1 (Figure 3). Accordingly, oleacein treatment dose-dependently suppressed the demethylase activity of LSD1 in vitro, with a mean IC_50_ of ~2.5 μmol/L.

### 3.4. Oleacein Suppresses LSD1-Targeted Enhancer-Driven Activation of the Stemness Transcription Factor SOX2 in Stem-Like Cells

LSD1-targeted inhibitors are known to differentially target pluripotent cancer cells, including teratocarcinoma, embryonic carcinoma, and seminoma or embryonic stem cells that express the core stem cell transcriptional factor SOX2, while having minimal growth-inhibitory effects on non-pluripotent or normal somatic cells [35,36,37,38,39,40,41,42]. Because this phenomenon can be explained in terms of the ability of LSD1 to specifically target a distal enhancer that controls *SOX2* expression in stem-like cells, we hypothesized that oleacein should prevent the overexpression of *SOX2* in breast cancer stem cells (CSC) via LSD1-regulated re-activation of this enhancer in the *SOX2* promoter [33,35,36,37,38,39,40,41,42]. As expected, when we transfected BT-474 breast cancer cells with a luciferase reporter vector containing the *SOX2* distal enhancer region centered between −3444 and −3833 [34], we observed a robust induction (>9-fold in average) in reporter activity in mammosphere cultures when compared with adherent differentiated control cultures (Figure 4). Of note, the LSD1-targeted, enhancer-driven transcriptional re-activation of *SOX2* differentially occurring in mammosphere cultures was fully suppressed, in a dose-dependent manner, following exposure to graded concentrations of oleacein (Figure 4).

We then took advantage of two different SOX2-overexpressing cellular models to substantiate the ability of oleacein to specifically target LSD1-driven stemness-related SOX2 expression. We first employed induced pluripotent stem (iPS) cells in which SOX2 is robustly transcribed from the LSD1-targeted distal enhancer to maintain pluripotency [44,45,46]. We then employed MCF-7 cells engineered to drive stable expression of SOX2 through transfer of the mCitrinie-P2A-Sox2 lentiviral vector [33,47,48], in which SOX2 overexpression is under the transcriptional control of the human phosphoglycerate kinase promoter. Oleacein treatment dose-dependently suppressed SOX2 protein expression in iPS cells; however, oleacein failed to modify ectopic SOX2 overexpression driven by a constitutive promoter in MCF-7/SOX2 cells (Figure 5). Transcriptional analyses confirmed the ability of oleacein to exclusively down-regulate endogenous *SOX2* expression in iPS cells but not in MCF-7/SOX2 cells expressing an exogenous *SOX2* transgene (Figure 5).

## 4. Discussion

In the present study, we used several approaches to identify the lysine-specific histone demethylase LSD1 as a novel target for oleacein, a characteristic bioactive biophenol-secoiridoid in EVOO.

First, we provide computational evidence supporting a catalytic/enzymatic inhibitory mechanism of action likely involving highly potent and direct targeting of the FAD cofactor cavity of LSD1 at physiological concentrations of oleacein. Because the interaction of LSD1 with RCOR1/CoREST allows the formation of an H3-tail binding-site recognizing a 20-amino-acid portion of the histone H3 N-terminal region [49,50], we cannot exclude the possibility that higher concentrations of oleacein might additionally target LSD1 via a scaffolding/structural mechanism involving disruption of the chromatin-binding activity of the LSD1/RCOR1/CoREST complex. Upon successful experimental validation of computational predictions, such information should be useful in the development of new secoiridoids-based structural inhibitors of LSD1.

Second, using an in vitro system with recombinant human LSD1 and a short peptide of methylated H3, we found that oleacein inhibits the enzymatic activity of LSD1 in a dose-dependent fashion, with an IC_50_ value (2.5 μmol/L) that can be achieved physiologically given that oleacein can be absorbed into systemic circulation, where it reaches concentrations up to 20 μmol/L [6].

Third, a pharmacological blocker of LSD1 is expected to promote methylation of H3K4 and H3K9 within the regulatory (enhancer) region of the *SOX2* promoter in pluripotent stem-like cells [40]. Accordingly, over the same concentration range that inhibited LSD1 activity in vitro, oleacein treatment efficiently blocked the re-activation of *SOX2* that is known to be controlled by the binding of LSD1 to the *SOX2* distal enhancer differentially occurring in tumor cells with CSC-like properties. Because oleacein was previously shown to exhibit strong suppressive effects against tumor-initiating CSC [10]—a sub-group of cancer cells having an enhanced ability to initiate tumorigenesis and drive therapeutic relapse—and considering the capacity of LSD1 to confer stem-cell-like traits to cancer cells [51,52], it is conceivable that some of the anti-CSC effects of oleacein could occur through LSD1 inhibition.

Given the central role of LSD1 for sensing and integrating energetic information into the epigenome [18,19,20,21,22,23,24,25,53,54], the LSD1-targeted inhibitory activity of oleacein might represent a novel mechanism through which EVOO phenolics impact the crosstalk between the metabolism and epigenetic gene regulation to exert health-promoting effects in a wide variety of human diseases, including obesity-associated diseases, neurological disorders, and cancer (Figure 6).

## 5. Conclusions

Overall, our study implicates a novel mechanism of action for the EVOO bioactive, phenolic oleacein, adding to the existing list of natural polyphenols capable of inhibiting LSD1, and laying the groundwork for the design of new secoiridoids-based structural inhibitors of LSD1.

## Figures and Tables

**Figure 1 nutrients-11-01656-f001:**
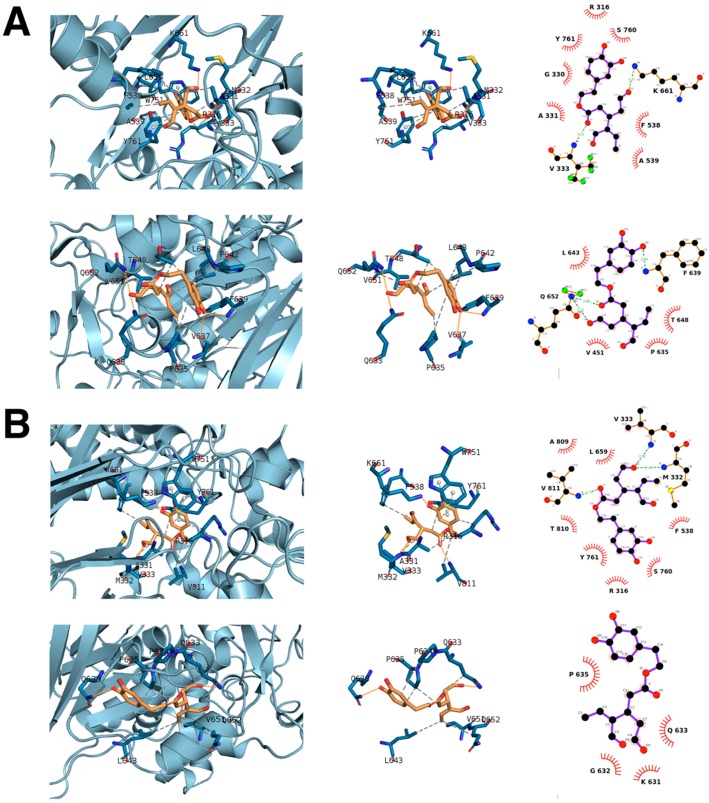
Structural analysis of oleacein binding to LSD1. Left panels show in sticks the interaction residues between oleacein and LSD1 (PDB: 2IW5) represented in the cartoon using PLIP. Middle panels show all the residues involved in oleacein binding using PLIP. Hydrogen bond interactions are represented by orange dashed lines; salt bridges are represented by yellow dashed lines and charge centers by yellow spheres; cation-π interactions are represented by blue dashed lines and white spheres represent the center of the aromatic ring. Right panels show all the residues involved in oleacein binding using LigPlot^+^. Large spoked arcs represent protein residues making non-bonded contacts with the ligand, whereas the small spoked arcs around atoms correspond to those of the ligand involved in these interactions; green dashed lines represent hydrogen bonds and their length is highlighted in green. Differences in the residues observed between panels are easily explained since each software tool incorporates different distances’ schema that ultimately define which residues or interactions are shown. Thus, the combination of both observations gives a wider and more complete view of oleacein binding. The residue numbers shown correspond to the original protein data bank (PDB) file numbering. (**A**), top: rigid docking; (**A**), bottom: blind docking; (**B**), top: docking molecular dynamics; (**B**), bottom: blind docking molecular dynamics.

**Figure 2 nutrients-11-01656-f002:**
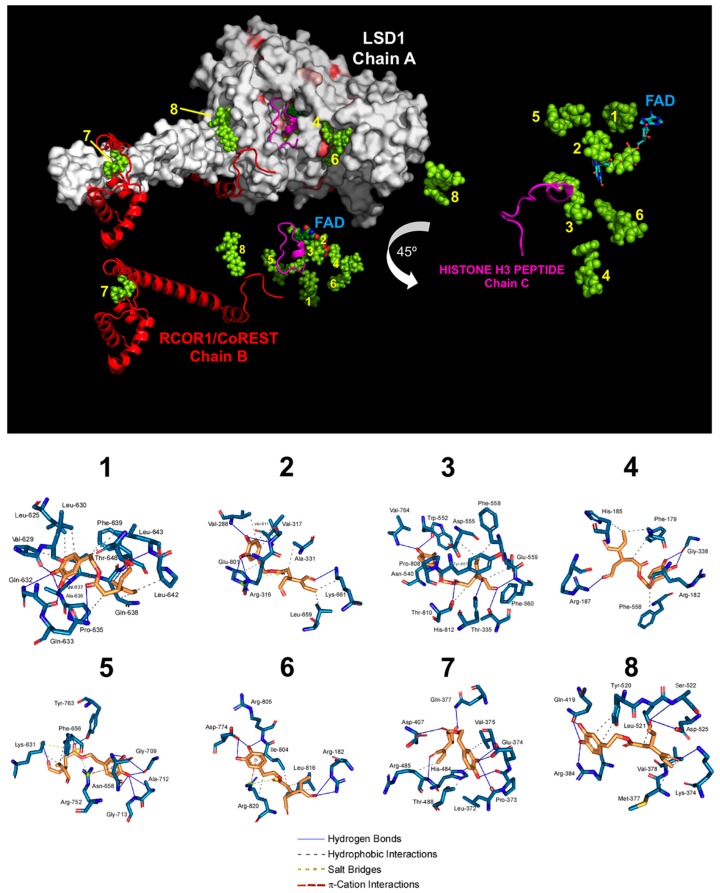
Binding mode of oleacein to LSD1. *Top.* The backbone of LSD1 (chain A)/REST corepressor 1 (chain B)/histone H3 peptide (chain C) heterotrimeric complex is shown (PDB: 2X0L). For each cluster (#1 to #8) of the docked oleacein (green color), only the molecule (spheres) with better binding energy is shown. The molecular docking was performed using the A and B chains in the absence of FAD and histone H3 peptide; however, the clusters of docked oleacein are shown superimposed on the position that would occupy both the FAD and the histone H3 peptide. The cluster number is also indicated. The detail on the right shows the peptide histone H3 (chain C, backbone as cartoon and side chains as sticks) and FAD (represented as spheres and with the blue carbons), including the best pose of oleacein docked in each cluster and the situation of the histone H3 peptide. *Bottom.* The detailed map of the molecular interactions of oleacein in each cluster is detailed (see also Table 2). Each inset shows the detailed interactions of each compound docked to the LSD1 heterodimer, indicating the participating amino acids involved in the interaction and the type of interaction (hydrogen bonds, hydrophilic interactions, salt bridges, ∏-stacking, etc).

**Figure 3 nutrients-11-01656-f003:**
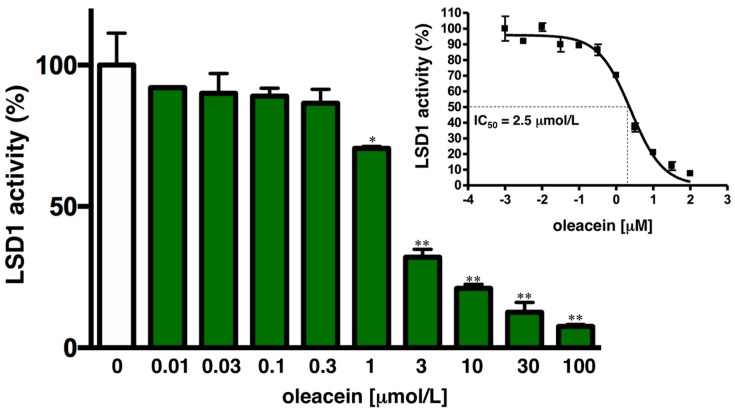
Oleacein inhibits LSD1 activity. Dose–response curves of LSD1 demethylation activity were created by plotting AlphaScreen signals as the function of oleacein concentration. Data (mean values and S.D. bars) are representative of two independent experiments (* *p* < 0.05, ** *p* < 0.005).

**Figure 4 nutrients-11-01656-f004:**
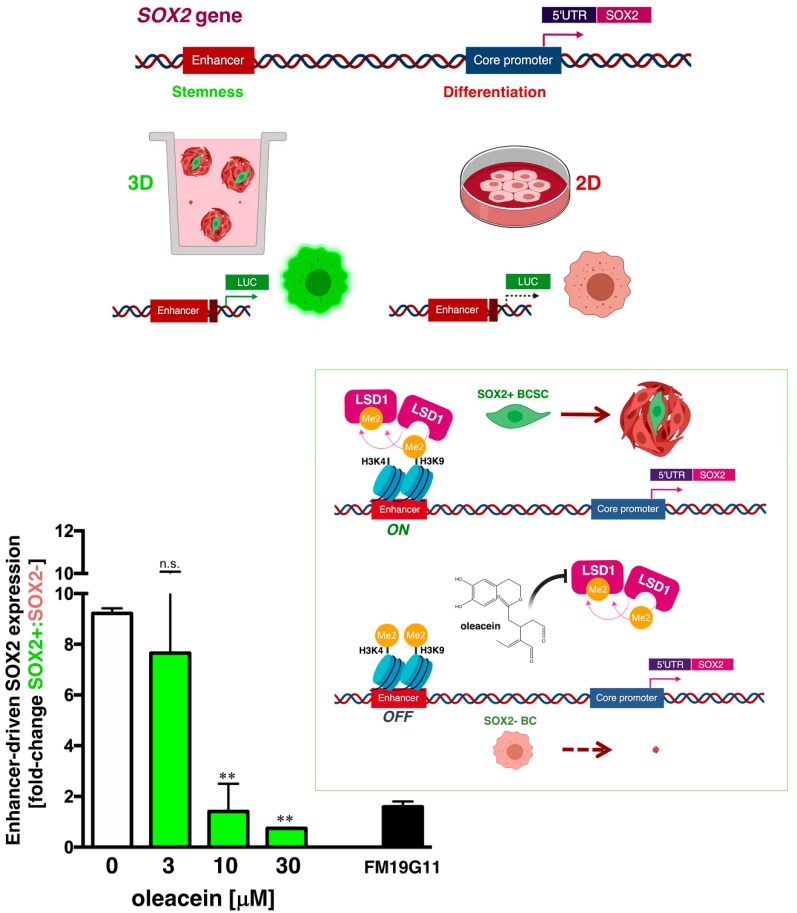
Oleacein suppresses LSD1-dependent, enhancer-driven expression of *SOX2* in cancer stem cells. Top. Schematic representation of *SOX2* promoter structure indicating the proximal core promoter region and the location of the distal enhancer, which is induced exclusively upon CSC-driven mammosphere formation but not in cell-adherent differentiating conditions. Bottom. Results are expressed as fold-induction of mammosphere culture-associated *SOX2* reporter activity over adherent culture control in the absence or presence of graded concentrations of oleacein. The results are expressed as percentages means (columns) ± SD (bars). ** *p* < 0.005, statistically significant differences from the untreated (control) group. FM19G11, an epigenetic repressor of key genes involved in stemness, including *SOX2* [43], was employed as a positive control. Inset. The use of fluorescence protein expression-based transcriptional reporters for activation of the LSD1-regulated enhancer element of the *SOX2* gene promoter can specifically identify cells with tumor-initiating activity; compounds that would be capable of impeding SOX2 activation might be viewed as valuable candidates for drugs aimed to target cancer stem cells (CSC). Our findings unravel the ability of oleacein to inhibit an LSD1-targeted distal enhancer that specifically controls the expression of the stem cell transcription factor SOX2 in pluripotent stem cells, thereby suppressing the re-activation of SOX2 exclusively occurring in mammosphere-initiating breast CSC [33]. The ability of oleacein to efficiently and specifically target the on/off LSD1-driven SOX2 regulatory process that provides higher tumorigenic potential to cells with an epigenetically acquired CSC phenotype confirms and extends previous findings from our laboratory showing the strong anti-CSC activity of oleacein [10].

**Figure 5 nutrients-11-01656-f005:**
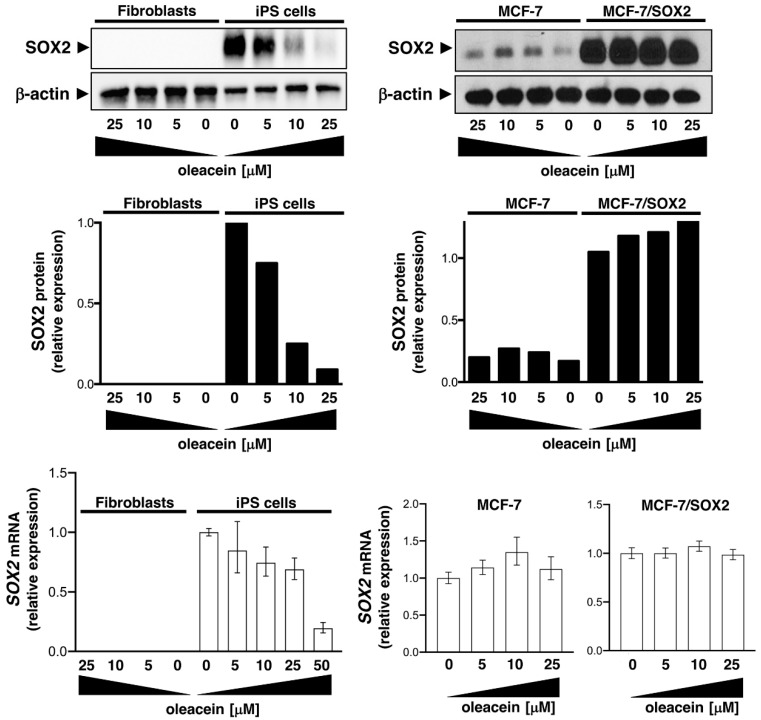
Oleacein suppresses stemness-related SOX2 protein expression. Western blotting analyses of SOX2 protein levels in human fibroblasts, induced pluripotent stem (iPS) cells, MCF-7, and MCF-7/SOX2 cells cultured in the absence or presence of graded concentrations of oleacein. β-actin expression was employed as an internal loading control. Images are representative of two independent experiments. Total RNA from human fibroblasts, iPS cells, MCF-7, and MCF-7/SOX2 cells cultured in the absence or presence of graded concentrations of oleacein was characterized in technical triplicates for the relative abundance of *SOX2* using RT-PCR analysis. The transcript abundance was calculated using the delta Ct method and presented as fold-change versus basal expression in untreated cells.

**Figure 6 nutrients-11-01656-f006:**
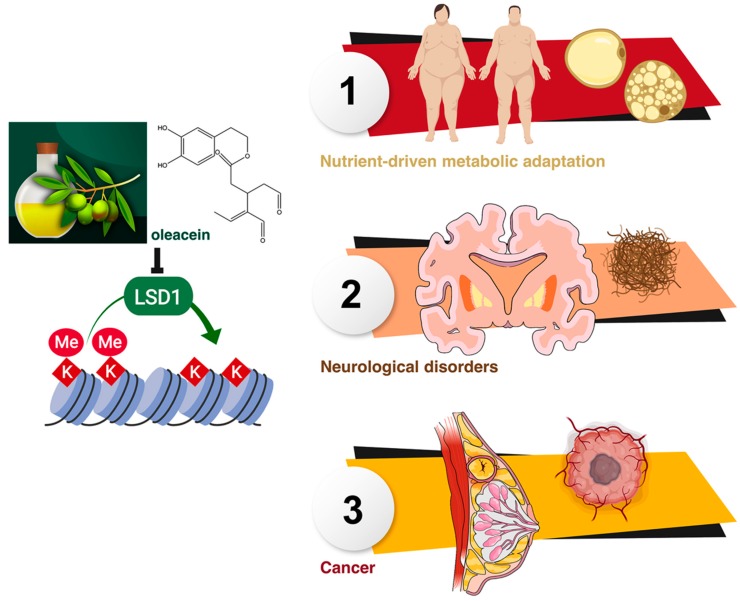
Oleacein-targeted LSD1: A new health-promoting mechanism of extra virgin olive oil (EVOO) phenolics.

**Table 1 nutrients-11-01656-t001:** Binding energies of oleacein against LSD1.

**PDB**	**Ligand**	**Chain**	**Docking Energy (kcal/mol) ^a^**	**MM/GBSA (kcal/mol) ^a^**
2IW5	FAD	A–B	−7.2/−7.5 ^b^	−39.9541
**PDB**	**Chain**		**Docking Energy (kcal/mol) ^c^**	**MM/GBSA (kcal/mol) ^c^**
2IW5	A–B		−6.1/−6.1 ^b^	−8.7173

^a^ Classical rigid docking and molecular mechanics-generalized born surface area (MM/GBSA)-based energy rescoring calculation over molecular dynamics simulation of oleacein against the FAD-binding cavity of LSD1. The more negative the binding energy, the more plausible the interaction. ^b^ Each calculation was performed twice to avoid false positives. ^c^ Blind docking and MM/GBSA-based energy rescoring calculation over molecular dynamics simulation of oleacein against the best second/third selected cavity of LSD1. The more negative the binding energy, the more plausible the interaction.

**Table 2 nutrients-11-01656-t002:** Interactions between oleacein and the LSD1-CoREST-histone H3 complex (PDB: 2X0L).

Cluster Number	∆G, (kcal/mol)	Dissoc. Constant, (µM)	Members	MM/PBSA, Solvation Binding Energy, (kcal/mol)	Residues of the Receptor That Contact Oleacein
1	−9.63	0.08779	13%	−66.844	Leu-625, Val-629, Leu-630, Lys-631, Gln-632, Gln-633, Pro-635, Ala-636, Val-637, Gln-638, Phe-639, Pro-642, Leu-643, Thr-648, Val-651 (chain A)
2	−9.58	0.09509	14%	−155.861	Gly-287, Val-288, Gly-314, Gly-315, Arg-316, Val-317, Leu-329, Gly-330, Ala-331, Phe-538, Leu-659, Asn-660, Lys-661, Trp-751, Ser-760, Tyr-761, Glu-801, Ala-809, Thr-810, Val-811, Ala-814 (chain A)
3	−9.33	0.14574	5%	−41.285	Thr-335, Ala-539, Asn-540, Leu-547, Trp-552, Asp-553, Asp-555, Asp-556, Phe-558, Glu-559, Phe-560, His-564, Ser-762, Tyr-763, Val-764, Tyr-773, Asn-806, Tyr-807, Pro-808, Ala-809, Thr-810, His-812 (chain A)
4	−8.30	0.82642	17%	−10.762	Pro-171, Glu-175, Ala-178, Phe-179, Arg-182, Leu-183, Pro-184, His-185, Asp-186, Arg-187, Gly-338, Gly-339, Asp-557, Phe-558, Glu-559, Phe-560, Thr-561, Tyr-807 (chain A)
5	−7.85	1.75000	6%	−6.730	Ala-541, Thr-542, Pro-543, Thr-546, Leu-627, Gly-628, Lys-631, Gly-655, Phe-656, Gly-657, Asn-658, Gly-709, Ala-712, Gly-713, Arg-752, Arg-758, Gly-759, Tyr-763 (chain A)
6	−7.62	2.61000	5%	−49.536	Arg-182, Gln-191, His-259, Leu-261, Tyr-773, Asp-774, Ala-777, Thr-803, Ile-804, Arg-805, Asn-806, Leu-816, Ser-817, Leu-819, Arg-820, Arg-824 (chain A)
7	−7.28	7.18000	25%	−40.682	Lys-481, His-484, Arg-485, Thr-488 (chain A) and Leu-372, Pro-373, Glu-374, Val-375, Ile- 376, Gln-377, Asp-407, Val-408, Gly-410 (chain B)
8	−6.65	13.32000	10%	−30.051	Lys-374, Met-377, Val-378, Glu-381, Arg-384, Gln-419, His-422, Val-423, Ser-517, Val-519, Tyr-520, Leu-521, Ser-522, Asp-525 (chain A) and Gly-313, Met-314, Phe-315 (chain B)

For the best-docked oleacein molecule of each cluster, the Gibbs free energy, dissociation constant, solvation binding energy (MM/PBSA calculated from 10 ns of molecular dynamics), and the number of molecules members (as %) of each cluster are shown.

## Data Availability

All data generated or analyzed during this study are included in this published article.

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
