# Peer review of "Extra Virgin Olive Oil Contains a Phenolic Inhibitor of the Histone Demethylase LSD1/KDM1A"

_nutrients, 2019, doi:10.3390/nu11071656_

Reviewer 1 Report

The article entitled “Extra virgin olive oil contains a phenolic inhibitor of the histone demethylase LSD1/KDM1A” describes the oleacein effect on LSD1. Authors described the inhibitory effect that this EVOO compound could exert on LSD1, and remarks the possible role as a new secoriridoid-based LSD1 inhibitor. In my opinion, this work is an original research that could clarify the role attributed to EVOO in several diseases. But, there is some comments that author has to take into account:  

KEYWORDS:

·       Olive oil and LSD1 words should be eliminated, due to their inclusion in the main title.

METHODS:

·       Line 107. “the samples were measured in the alphaScreen…” à Please, replace “in” for “by”.

·       Line 136. “in the absence or presence of graded concentrations” à Please could you specify concentrations, for example “in the absence or presence of ranged concentration from X to Y”.

·       Line 144. “graded concentrations” à same comment as above.

RESULTS:

·       Fig 1 and Fig 2. This comment is more for edition, that for authors. But please, editors should consider to maximize as far as they can size of fig 1 and 2. Because it is difficult to read numbers and visualize structures. 

·       Fig 3. Why S.D. bar at 0.01 µmol/L is inexistent?

·       Fig 4. It should be picture by separate, in two figures to improve the scheme of the right fig 4A.

Author Response

We would like to thank this reviewer for the positive remarks on our manuscript.  Below, we have addressed all the comments and questions raised by the reviewer point-by-point. We have made changes in the text accordingly (in red). The original reviewers’ comments are italicized and our responses to these comments follow (in bold)

KEYWORDS:

Olive oil and LSD1 words should be eliminated, due to their inclusion in the main title.

Done!

METHODS:

Line 107. “the samples were measured in the alphaScreen…” Please, replace “in” for “by”

Because this sentences refers to the alphaScreen reader rather than the technique, we have now replaced “in” by “on an” (Line 107)

Line 136. “in the absence or presence of graded concentrations”. Please could you specify concentrations, for example “in the absence or presence of ranged concentration from X to Y”.

Done!

Line 136. “graded concentrations”. Same comment as above

Done!

RESULTS:

Fig 1 and Fig 2. This comment is more for edition, that for authors. But please, editors should consider to maximize as far as they can size of fig 1 and 2. Because it is difficult to read numbers and visualize structures.

Fig 3. Why S.D. bar at 0.01 umol/L is inexistent?

The sample standard deviation of this data set was zero because all of its values were identical.

Fig 4. It should be picture for separate, in two figures to improve the scheme of the right fig. 4A.

Figure 4 has now been split into two as per the reviewer's suggestion, namely Figure 4 (previous Fig. 4A) and Figure 5 (previous Fig. 4B).

Reviewer 2 Report

The manuscript by Menendez group describes new findings that oleacein from EVOO is a potential inhibitor to target LSD1. They used different computational approaches to identify the potential binding mode of oleacein on LSD1 or related complexes and indicated that the FAD-site would be the potential binding site. The authors further functionally validated the in silico predictions by enzymatic assays and different cellular assays. Overall, I feel the study is interesting and very well done. It highlights the beneficial ingredient of EVOO in our diet and for potential biomedical use. The paper is well written.

Minor comments:

1) It is always important to confirm the target engagement. As suggested by the in silico prediction, at high concentrations, oleacein might target LSD1 with histone H3 and the RCOR1/CoREST. Can the author validate this by bioassays?

2) In Figure 1, it would be better if the authors can present the protein 3D structures at the same perspective and only show the different binding poses of oleacein on the protein among using rigid, blind, and MD dockings. That would help the readers to better understand the results from different docking methods.

3) there is a mislabel of footnote c in table 1.

Author Response

We would like to thank this reviewer for the positive remarks on our manuscript.  Below, we have have tried to address some of her/his comments and questions point-by-point. We have made changes in the text accordingly (in red). The original reviewers’ comments are italicized and our responses to these comments follow (in bold).

Minor comments:

It is always important to confirm the target engagement. As suggested by the in silico prediction, at high concentrations, oleacein might target LSD1 with histone H3 and the RCOR1/CoREST. Can the author validate this by bioassays?

We acknowledge the reviewer’s suggestion. However, we honestly believe that such validation is beyond the scope of this manuscript. Nevertheless, we acknowledge in the discussion section that such experimental validation of computational predictions should be useful in the development of new secoiridoids-based structural inhibitors of LSD1.

Lines 346-348

Upon successful experimental validation of computational predictions, such information should be useful in the development of new new secoiridoids-based structural inhibitors of LSD1.

In Figure 1, it would be better if the authors can present the protein 3D structures at the same perspective and only show the different binding poses of oleacein on the protein among using rigid, blind, and MD dockings. That would help the readers to better understand the results from different docking methods.

We appreciate the reviewer’s suggestion. Because date in Figure 1 arise from a computational screening performed using proprietary software from an artificial intelligence-based computational company (i.e., Mind the Byte), they solely aim to present the notion that LSD1 was a predicted putative target for oleacein. Figure 2, using a different PDB for LSD1, validates the prediction using publicly available software approaches and already present the protein 3D structure at the same perspective employed to generate the binding mode data included in Table 2.

Alternatively, all the figure panels from Figure 1 can be moved to a supplementary File to keep the complexity of this brief communication to a minimum.

There is a mislabel of footnote c in table 1.

Thank you for pointing it out. Corrected!